# LATENT WEIGHT DIFFUSION:
# GENERATING POLICIES FROM TRAJECTORIES

## ABSTRACT

With the increasing availability of open-source robotic data, imitation learning has emerged as a viable approach for both robot manipulation and locomotion. Currently, large generalized policies are trained to predict controls or trajectories using diffusion models, which have the desirable property of learning multimodal action distributions. However, generalizability comes with a cost — namely, larger model size and slower inference. Further, there is a known trade-off between performance and action horizon for Diffusion Policy (i.e., diffusing trajectories): fewer diffusion queries accumulate greater trajectory tracking errors. Thus, it is common practice to run these models at high inference frequency, subject to robot computational constraints.

To address these limitations, we propose Latent Weight Diffusion (LWD), a method that uses diffusion to learn a distribution over policies for robotic tasks, rather than over trajectories. Our approach encodes demonstration trajectories into a latent space and then decodes them into policies using a hypernetwork. We employ a diffusion denoising model within this latent space to learn its distribution. We demonstrate that LWD can reconstruct the behaviors of the original policies that generated the trajectory dataset. LWD offers the benefits of considerably smaller policy networks during inference and requires fewer diffusion model queries. When tested on the Metaworld MT10 benchmark, LWD achieves a higher success rate compared to a vanilla multi-task policy, while using models up to ~18x smaller during inference. Additionally, since LWD generates closed-loop policies, we show that it outperforms Diffusion Policy in long action horizon settings, with reduced diffusion queries during rollout.

## 1 INTRODUCTION

The recent increase in open-source robotic data has made imitation learning an attractive prospect to solve robot manipulation and locomotion tasks (Collaboration et al., 2023; Peng et al., 2020). While traditional supervised learning methods like Behavioral Cloning (Florence et al., 2022) and transformer-based models such as RT-1 (Brohan et al., 2022) have demonstrated some success, they are unable to capture the multimodal nature of robotic action distributions (e.g., while avoiding an obstacle in a navigation task with two optimal actions in opposing directions 'turn left' and 'turn right', the learned action 'go straight' is a suboptimal average of the two). Recently, diffusion-based methods have emerged as a promising alternative for robot control (Tan et al., 2024), offering the advantages of continuous outputs and the capacity to learn multimodal action distributions.

Inspired by the success of latent diffusion in vision (Rombach et al., 2022b) and language (Lovelace et al., 2024), we explore it here for robotics. We introduce a novel method that utilizes diffusion models to learn a distribution of policies for robotic tasks from demonstration data. Unlike existing robotics approaches focusing on trajectory diffusion (Chi et al., 2024), our method Latent Weight Diffusion (LWD) diffuses neural network weights to generate policies. This is achieved by encoding demonstration trajectories into a latent space, employing a diffusion denoising model to learn the distribution of latents within this space, followed by decoding the latent representations into executable policies using a hypernetwork (Ha et al., 2016).

LWD provides several benefits. First, the distribution modeling capabilities of diffusion models allow it to capture complex behavior distributions. Second, generating the parameters of a neu-

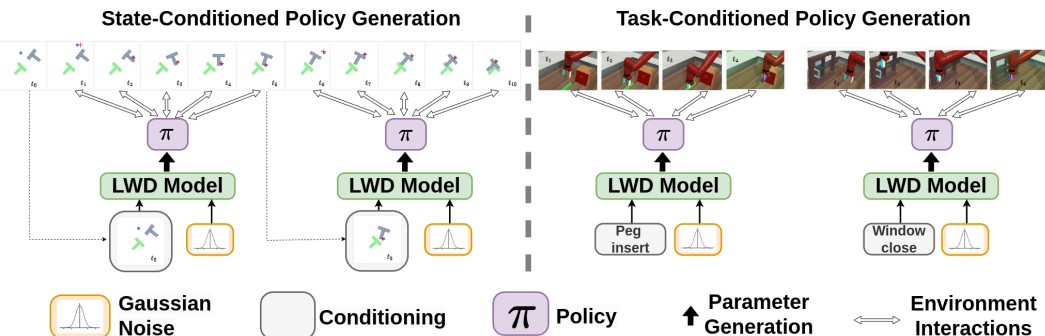

Figure 1: **Latent Weight Diffusion (LWD)** generates policies from heterogeneous trajectory data. With state-conditioned policy generation, the diffusion model can run inference at a lower frequency. With task-conditioned policy generation, the generated policies can be small yet maintain task-specific performance. Demonstrations of this work can be found on the project website: https://sites.google.com/view/iclr2024submission/home

ral network policy enables them to be run as closed-loop controllers. Since closed-loop control is less susceptible to trajectory tracking errors than open-loop control, LWD allows for longer action horizons than trajectory generation methods. This has the additional advantage of reducing calls to an expensive diffusion model. Third, the LWD model can be conditioned on task identifiers to generate task-specific policies. This provides a unique advantage of maintaining multi-task generalization in the diffusion model and task-specific performance in the generated policy. Since the generated policy is task-specific, it has fewer parameters than a generalist multi-task agent during inference. These advantages demonstrate the effectiveness of latent diffusion in learning policies from trajectory data, leading to the following performance gains.

1. **Policy Diversity**: LWD learns a diverse policy distribution from a trajectory dataset. On the D4RL benchmark, when trained on a trajectory mixture from three behaviors, LWD accurately generates policies capturing the behavior distributions of the original policies.

2. **Closed-loop Control**: LWD generates closed-loop policies enabling reactivity to environmental changes. This enables policies generated by LWD to run for 4X longer action horizons, compared to a Diffusion Policy model for the same change in performance, when evaluated on the PushT task.

3. **Small Policy Size**: The burden of generalization is borne by the diffusion model instead of the inference policy, allowing the generated policy to have fewer parameters, without compromising multi-task performance. On the Metaworld MT10 suite of tasks, LWD shows similar multi-task performance as a BC baseline that is up to 18X larger.

## 2 RELATED WORK

### 2.1 IMITATION LEARNING FOR ROBOTICS

With the availability of vast swaths of open-source robotic data, imitation learning has emerged as a viable approach for robot control. Original imitation learning approaches were simple – behavioral cloning agents that learned to predict controls or trajectories from expert demonstrations. With the advent of transformer-based methods, imitation learning has grown in popularity. Methods like PerAct (Shridhar et al., 2022) and RT-1 (Brohan et al., 2022) perform well on various tasks. Brohan et al. (2023) showed that VLMs can be combined with robot demonstrations to solve tasks by directly interpreting the token output as actions. Collaboration et al. (2023) showed that transformer-based methods can be used to learn from demonstrations across different embodiments. Object-aware policies like Heravi et al. (2022) have been shown to improve performance for visuomotor tasks.

### 2.2 DIFFUSION

Diffusion models have emerged as a leading approach in image generation, with Denoising Diffusion Probabilistic Models (DDPM) being a prominent class of generative models (Ho et al., 2020).

These models generate images by progressively denoising samples drawn from an isotropic Gaussian distribution. Furthermore, Rombach et al. (2022a) demonstrated that diffusion can be effectively applied in the latent space of a pre-trained Variational Autoencoder. Recently, diffusion-based methods have shown promise in solving robotic tasks. The seminal work Chi et al. (2024) showed that diffusion models can be used to learn multimodal action distributions for a task by diffusing trajectories for control up to a defined action horizon. Urain et al. (2022); Luo et al. (2024); Carvalho et al. showed that diffusion models can be used to learn smooth cost functions for the joint optimization of grasp and motion plans. For long horizon skills, Mishra et al. (2023) showed how to use diffusion to chain skills together to solve a larger task. Diffusion models have been used to formulate policies to control a quadruped robot Huang et al. (2024), although the length of the trajectories diffused was still relatively short, therefore requiring a higher diffusion inference frequency. Tan et al. (2024) showed that latent diffusion could be applied to multi-task manipulation action trajectory generation. A key limitation of using diffusion models to generate trajectories is the degradation in performance for longer action horizons, which is caused by both modeling and trajectory tracking errors.

### 2.3 HYPERNETWORKS

Hypernetworks, introduced by Ha et al. (2016), are neural networks that can estimate the weights of a secondary network. Following their inception, they have been extended and applied in multiple settings. In meta-learning, Bertinetto et al. (2016) proposed a model where a learner network predicts the parameters of another network for one-shot learning tasks, sharing conceptual similarities with hypernetworks. The concept of dynamically generating network parameters is also related to Dynamic Filter Networks by Jia et al. (2016), where filters are generated on the fly based on the input. This method aligns with the principles of hypernetworks, emphasizing adaptability and efficiency in processing varying inputs. It has also been shown that hypernetworks can be used for robot policy representation (Hegde et al., 2024).

### 2.4 MULTI-TASK LEARNING

Metaworld (Yu et al., 2020) and RLBench (James et al., 2019) are popular benchmarks for multi-task learning. Many recent transformer-based methods have shown good performance on various tasks, given task conditioning during training and testing, such as Shridhar et al. (2022) and the RT family of models. GNFactor (Ze et al., 2023) uses a generalizable neural feature field to learn a volumetric representation of the environment, which can be used to synthesize different views of the environment. Another way to solve multi-task learning is through modularity. Devin et al. (2016) showed how to split networks into modules that are task-specific and robot-specific. Naturally, the task-specific modules can be shared across robots, and the robot-specific modules can be shared across tasks on a robot, enabling the transfer of learned behaviors across tasks or robot embodiments.

## 3 PROBLEM FORMULATION & METHOD

This work builds on Hegde et al. (2023), which demonstrates the capacity of latent diffusion models to generate policies from a policy dataset while addressing its key limitation - the reliance on often unavailable policy datasets - by utilizing trajectory datasets instead. LWD employs a two-step process. A variational autoencoder (VAE) with a weak KL-regularization coefficient encodes trajectories into a latent space that can be decoded into a trajectory. A diffusion model learns the distribution of this latent space, enabling policy sampling from the learned distribution (see Figure 2).

### 3.1 LATENT POLICY REPRESENTATION

Consider $\pi(\cdot, \theta)$ as a stochastic policy, parameterized by $\theta$, that interacts with the environment and generates trajectories $\tau$. Suppose there exists a distribution of policy parameters $p(\theta)$, and we sample a policy parameter from this distribution and collect a trajectory for this sampled policy parameter. This process of sampling the policy parameter is done for all trajectories collected. We assume that for a given $\theta$, $a_t = \pi(s_t, \theta) + e$, where $e$ is normally distributed around 0. i.e., $a_t \sim \mathcal{N}(\pi(s_t, \theta), \sigma^2)$

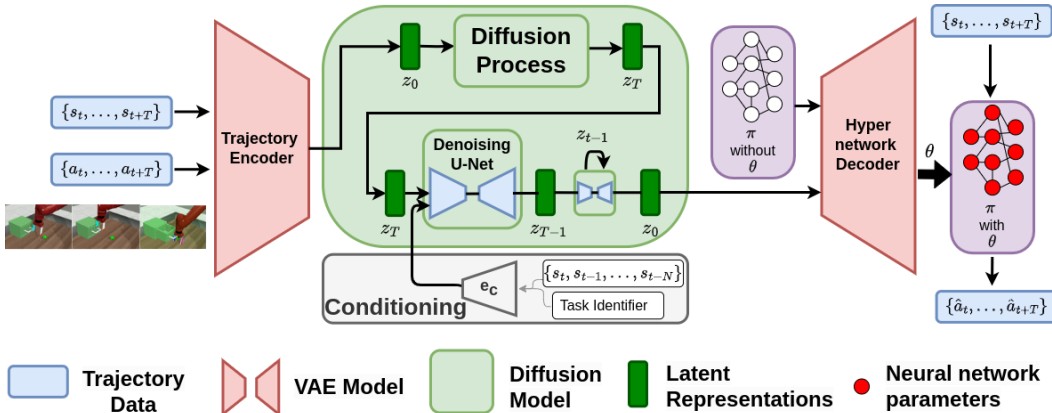

Figure 2: **LWD**: We first pre-train a VAE that variationally encodes trajectories to a latent space and then decodes it as policy parameters. Next, we train a conditional latent diffusion model to learn this latent distribution.

Our goal is to learn the distribution of policy parameters $p(\theta)$ that produced the trajectory dataset. We assume that a latent variable $z$ exists that contains information required to identify different policy behaviors. Since trajectories are generated using parameters $\theta$, we can use conditional independence, $p(\tau \mid z, \theta) = p(\tau \mid \theta)$. Considering that our dataset consists of trajectories, we want to maximize the likelihood of sampling $\tau$, therefore maximizing $\log p(\tau)$. We derive a modified version of the Evidence Lower Bound (ELBO) to incorporate $p(\theta)$ as shown below. The complete derivation is shown in Appendix A

$$mELBO = \sum_{t=1}^{T} \left[ \mathbb{E}_{q(z|\tau)} \left[ \mathbb{E}_{p(\theta|z)} \left[ \log p(a_t \mid s_t, \theta) \right] \right] \right] - \mathrm{KL}(q(z \mid \tau) \parallel p(z)) \tag{1}$$

## 3.2 VARIATIONAL AUTOENCODER FOR POLICIES

Since we now have a modified ELBO objective, we shall now try to approximate its components with a variational autoencoder. Let $\phi_{enc}$ be the parameters of the VAE encoder that variationally maps trajectories to $z$, and $\phi_{dec}$ be the parameters of the VAE decoder. We assume the latent $z$ is distributed with mean zero and unit variance. We construct the VAE decoder to approximate $p(\theta \mid z)$ with $p_{\phi_{dec}}(\theta \mid z)$. Considering $a_t \sim \mathcal{N}(\pi(s_t, \theta), \sigma^2)$, we derive our VAE loss function as:

$$\mathcal{L}\left(\{s_t^k, a_t^k\}_{t=1}^{T} \mid \phi_{enc}, \phi_{dec}\right) = -\sum_{t=1}^{T} \mathbb{E}_{q_{\phi_{enc}}(z|\{s_t^k, a_t^k\}_{t=1}^{T})} \left[ (a_t^k - \pi(s_t^k, f_{\phi_{dec}}(z)))^2 \right]$$

$$- \beta_{kl} \sum_{i=1}^{\dim(z)} \left( \sigma_i^2 + \mu_i^2 - 1 - \log \sigma_i^2 \right) \tag{2}$$

where, $(\mu, \sigma) = f_{\phi_{enc}}(\{s_t^k, a_t^k\}_{t=1}^{T})$, $z \sim \mathcal{N}(\mu, \sigma)$, and $\beta_{kl}$ is the regularization weight. The complete derivation is shown in appendix B Since the decoder in the VAE outputs the parameter of a secondary network, we shall use a conditional Hypernetwork, specifically the one that was developed for continual learning by von Oswald et al. (2020).

## 3.3 POLICY DIFFUSION

In practice, we see that approximating $p(z) = \mathcal{N}(0, I)$ is suboptimal, and therefore we set $\beta_{kl}$ to a very small number $\sim (10^{-9}, 10^{-6})$. After training the VAE to maximize the objective provided in Equation 7 with this $\beta_{kl}$, we have access to this latent space $z$ and can train a diffusion model to learn its distribution $p(z)$. The diffusion process in the latent space involves gradually adding noise

to the latent variable $\mathbf{z}_0 = \mathcal{E}(\tau)$ over a sequence of time steps $t \in \{0, 1, \ldots, T\}$, where $\mathcal{E}$ is the learned trajectory encoder. This process can be described by a forward noising process:

$$\epsilon_t = q(z_t \mid z_{t-1}) = \mathcal{N}\left(z_t; z_{t-1}\sqrt{1 - \beta_t}, \beta_t \mathbf{I}\right) \tag{3}$$

making the forward process Markovian. The reverse or generative process $p_{\phi_{dif}}(z_T)$ reverts noise from an isotropic Gaussian into a sample $z_0$ in $q(\mathbf{z})$ and contains a similar Markov structure.

$$p_{\phi_{dif}}(z_0) = p(z_T) \prod_{t=1}^{T} p_{\phi_{dif}}(z_{t-1} \mid z_t), \quad p_{\phi_{dif}}(z_{t-1} \mid z_t) = \mathcal{N}\left(z_{t-1}; \mu_{\phi_{dif}}(z_t, t), \Sigma_{\phi_{dif}}(z_t, t)\right) \tag{4}$$

We can condition the latent denoising process on the current state and/or the task identifier $c$ of the policy required. Therefore the model shall be approximating $p_{\phi_{dif}}(z_{t-1} \mid z_t, c)$. After denoising for a given state and task identifier, we can convert the denoised latent to the required policy.

Therefore, to sample from $p(\theta)$, first sample z using the trained diffusion model $z \sim p_{\phi_{dif}}(z_0)$, and then apply the deterministic function $f_{\phi_{dec}}$ to the sampled $z$.

## 4 EXPERIMENTS

We first analyze the behavior reconstruction of LWD , followed by the effect of breaking down the trajectory into shorter snippets. Then, we perform an ablation over the size of LWD , showing that a larger model can mitigate the problems introduced by snipping trajectories. After this, we benchmark LWD on the MT10 suite of tasks in Metaworld, showcasing multitask advantages, as well as a benchmark on human-generated data in the PushT environment, showcasing its ability to run at longer action horizons. All our numerical results are shown across three seeds.

We focus on demonstrating results in state-based low-dimension observation spaces. Thus, the generated policies are always Multi-Layer Perceptrons (MLP) with 2 hidden layers with 256 neurons each. In the VAE, the encoder is a sequential network that flattens the trajectory and compresses it to a low-dimension latent space. The decoder is a conditional hypernetwork from the hypernettorch package (Ehret et al., 2021). For the diffusion model, we adapt the model provided by Rombach et al. (2022a). For all experiments the latent space is $\mathbb{R}^{256}$, the KL regularization weight $\beta_{kl} = 10^{-8}$, the learning rate is $10^{-4}$ with the Adam optimizer.

### 4.1 BEHAVIOR RECONSTRUCTION ANALYSIS

Here, we ask – how does LWD perform in reconstructing the behavior of the original policies that generated the trajectory dataset? Is it able to reproduce different behaviors for the same task?

First, we analyze the behavior reconstruction capability of different components of LWD . For this experiment, we use the D4RL (Fu et al., 2020) halfcheetah dataset. Each trajectory in this dataset has a length of 1000. We combine trajectory data from three original behavior policies provided in this dataset: expert, medium, and random. Following Batra et al. (2023), we track the foot contact timings of each trajectory as a metric for measuring behavior. For each behavior policy, we get 32 trajectories. These timings are normalized to the trajectory length and are shown in Figure 3. For each plot, the x-axis denotes the foot contact percentage of the front foot, while the y-axis denotes the foot contact percentage of the back foot. We first visualize the foot contact timings of the original policies in Figure 3a. Then, we train the VAE model on this dataset to embed our trajectories into a latent space. We then apply the hypernetwork decoder to generate policies from these latents. These policies are then executed on the halfcheetah environment, to create trajectories. We plot the foot contact timings of these generated policies in Figure 3b. We see that the VAE captures each of the original policy's foot contact distributions, therefore empirically showing that the assumption $p_{\phi_{dec}}(\theta \mid z) = \delta(\theta - f_{\phi_{dec}}(z))$ is reasonable. Then, we train a latent diffusion model conditioned on a behavior specifier (i.e., one task ID per behavior). In Figure 3c, we show the distribution of foot contact percentages of the policies generated by the behavior specifier conditioned diffusion model. We see that the diffusion model can learn the conditional latent distribution well, and the behavior

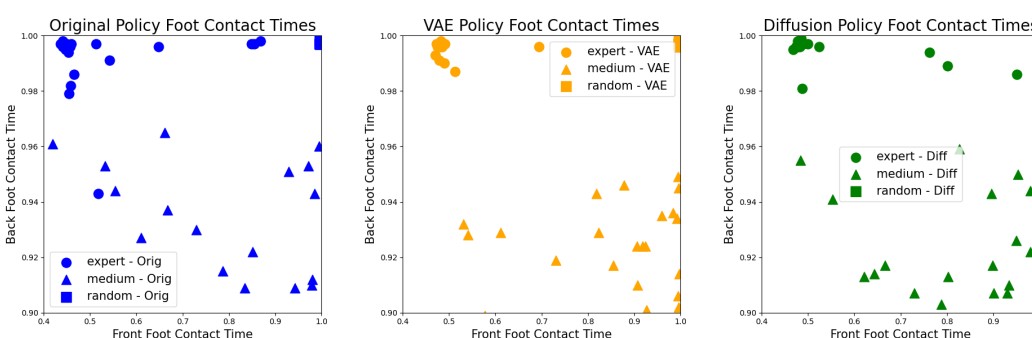

(a) Original policies that provide the trajectory dataset.

(b) VAE generated policies from trajectories.

(c) VAE + diffusion generated policies.

Figure 3: **Foot-contact times shown for various trajectories on the Half Cheetah task**. Figure 3a: We use foot contact times as the chosen metric to show different behaviors for the half cheetah run task by different policies. Figure 3b: Our VAE can embed these behaviors into a latent space and then reconstruct a policy from them with the same behavioral patterns. Figure 3c Our diffusion model then learns the distribution of this latent space. We train it with the task label as the conditioning.

distribution of the decoded policies of the sampled latent matches the original distribution. Note that to sample policies during inference, we do not need to encode trajectories; rather, we need to sample latents using the diffusion model and use the hypernetwork decoder to decode a policy from it.

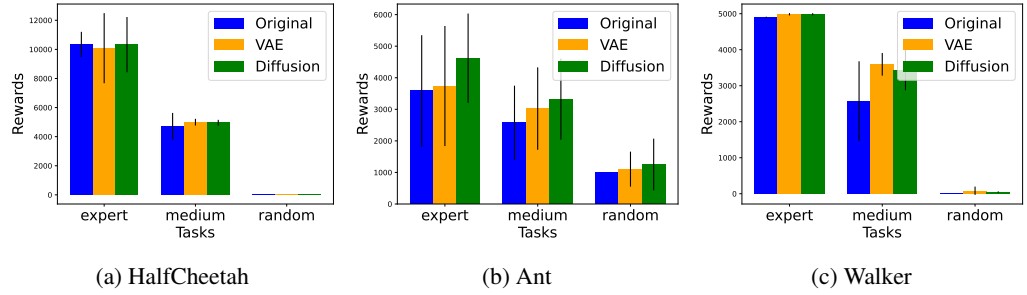

(a) HalfCheetah

(b) Ant

(c) Walker

Figure 4: **Reconstruction Rewards**: For each of the 3 environments shown above, the generated policy from trajectory decoded VAE and task-conditioned diffusion model, achieves similar total objective as the original policies. Each bar indicates the mean total objective obtained with error lines denoting the standard deviation.

Another way to analyze the behavior reconstruction capability of LWD is to compare the rewards obtained during a rollout. Figure 4 shows us the total objective obtained by the original, VAE-decoded, and diffusion-denoised policies. We see that the VAE-decoded and diffusion-generated policies achieve similar rewards to the original policy for each behavior.

Apart from these plots, we use Jensen-Shannon divergence to quantify the difference between two distributions of foot contact timings. Table 1 shows the JS divergence between the empirical distribution of the foot contact timings of the original policies and those generated by LWD. The lower this value is, the better. As a metric to capture the stochasticity in the policy and environment, we get the JS divergence between two successive sets of trajectories generated by the same original policy, which we shall denote SOS (Same as source). A policy having a JS divergence score lesser than this value indicates that that policy is indistinguishable from the original policy by behavior. As a baseline for this experiment, we train a large (5-layer, 512 neurons each) behavior-conditioned MLP on the same mixed dataset with MSE loss. We see that policies generated by LWD consistently achieve a lower JS divergence score than the MLP baseline for expert and medium behaviors. The random behavior is difficult to capture as the actions are almost Gaussian noise. Surprisingly, for the HalfCheetah environment, policies generated by LWD for expert and medium had lower scores than SOS, making it behaviorally indistinguishable from the original policy.

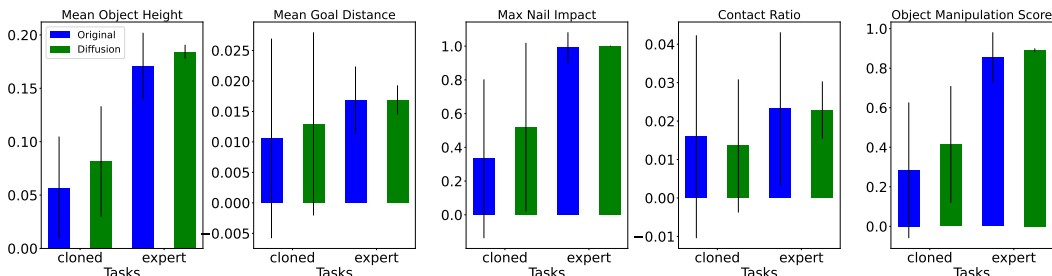

Figure 5: **Behavior Reconstruction for Manipulation**: We track these metrics on the Adroit hammer task, and the LWD-generated policy behaves similarly to the original policy

To verify the behavior reconstruction capabilities of LWD, we also experiment on the D4RL Adroit dataset (Rajeswaran et al., 2018). The task we choose is of tool use, where the agent must hammer a nail into a board. We utilize their 5000 expert and 5000 human-cloned trajectories, to train our LWD model. The implementation details are provided in appendix C.4. Then we evaluate the behavior of the original and the generated policy on the following metrics: **Mean object height** - Average height of the object during eval; **Alignment error (goal distance)** - Mean distance between the target and the final goal position; **Max nail impact** - Maximum value of the nail impact sensor during eval; **Contact ratio** - Fraction of time steps where the nail impact sensor value exceeds 0.8; **Object manipulation score** - Proportion of time steps where the object height exceeds 0.04 meters. From Figure 5, we can see that the policy generated by LWD behaves similarly to the original policy.

| Environment | Source Policy | Target Policy | | |
|---|---|---|---|---|
| | | SOS | MLP | LWD |
| Ant | Expert | $0.187 \pm 0.142$ | $1.272 \pm 0.911$ | $0.510 \pm 0.159$ |
| | Medium | $0.624 \pm 0.232$ | $1.907 \pm 0.202$ | $1.328 \pm 0.283$ |
| | Random | $1.277 \pm 1.708$ | $4.790 \pm 0.964$ | $8.859 \pm 0.792$ |
| HalfCheetah | Expert | $0.158 \pm 0.146$ | $2.810 \pm 1.139$ | $0.088 \pm 0.050$ |
| | Medium | $0.275 \pm 0.196$ | $0.692 \pm 0.787$ | $0.194 \pm 0.157$ |
| | Random | $0.0467 \pm 0.009$ | $0.11 \pm 0.009$ | $0.104 \pm 0.0187$ |
| Walker2D | Expert | $0.342 \pm 0.329$ | $2.879 \pm 1.493$ | $1.093 \pm 0.310$ |
| | Medium | $0.078 \pm 0.058$ | $0.165 \pm 0.126$ | $0.155 \pm 0.091$ |
| | Random | $0.080 \pm 0.004$ | $60.514 \pm 52.461$ | $2.776 \pm 1.260$ |

Table 1: **Behavior Reconstruction**: JS divergence between foot contact distributions from source and target policies. The lower the value, the better.

## 4.2 ENCODING TRAJECTORY SNIPPETS

Here, we ask the question – can LWD generate policies that are faithful to the original policies, even when provided with only a snippet of the trajectory data? For most robotics use cases, it is impossible to train on long trajectories due to the computational limitations of working with large batches of long trajectories. Therefore, we analyze the effect of sampling smaller sections of trajectories from the dataset. After training a VAE for the D4RL halfcheetah dataset on three policies (expert, medium, and random), we encode all the trajectories in the mixed dataset to the latent space. We then perform Principal Component Analysis (PCA) on this set of latents and select the first two principal components. Figure 6a shows us a visualization of this latent space. We see that the VAE has learned to encode the three sets of trajectories to be well separable. Next, we run the same experiment, but now we sample trajectory snippets of length 100 from the dataset instead of the full-length (1000) trajectories. Figure 6b shows us the PCA on the encoded latents of these trajectory snippets. We see that the separability is now harder in the latent space. Surprisingly, we noticed that after training our VAE on the snippets, the decoded policies from randomly snipped trajectories were still faithfully behaving like their original policies. We believe that this is because the halfcheetah env is a cyclic locomotion task, and all trajectory snippets have enough information to indicate its source policy.

To validate this hypothesis, we analyze our method on trajectory snippets for non-cyclic tasks. We choose the MT10 suite of tasks in Metaworld (Yu et al., 2020). We utilize the hand-crafted expert

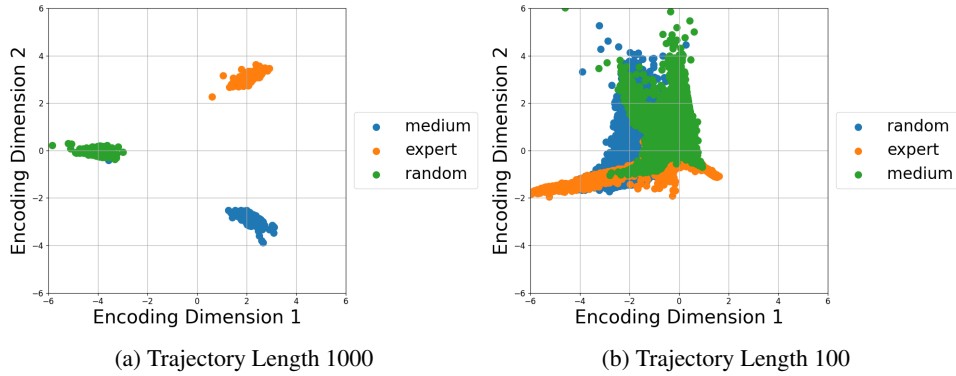

(a) Trajectory Length 1000

(b) Trajectory Length 100

Figure 6: **Effect of trajectory snipping** in HalfCheetah. Top two principal components of the latent.

policy for each of the tasks in MT10 to collect trajectory data. For each task, we collect 1000 trajectories of length 500.

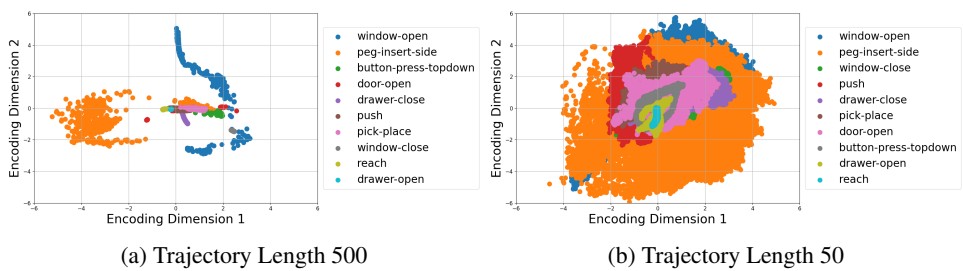

(a) Trajectory Length 500

(b) Trajectory Length 50

Figure 7: **Effect of trajectory snipping** in MT10. Top two principal components of the latent.

Figure 7a shows the principal components of the latents of the full trajectories in the dataset, and Figure 7b shows the same for the split trajectories. We can see that the separability of different tasks is much harder in this case. More dimensions of the PCA are shown in the Appendix D. Further, we noticed that the decoded policies from the trajectory snippets did not perform as well as the original policies - for the same decoder size as the half cheetah task. This validates our hypothesis that the snippets are unable to reproduce the original policy for non-cyclic tasks. To have the same degree of behavior reconstruction as the half-cheetah tasks, we need a larger decoder model. This is discussed next, in subsection 4.3.

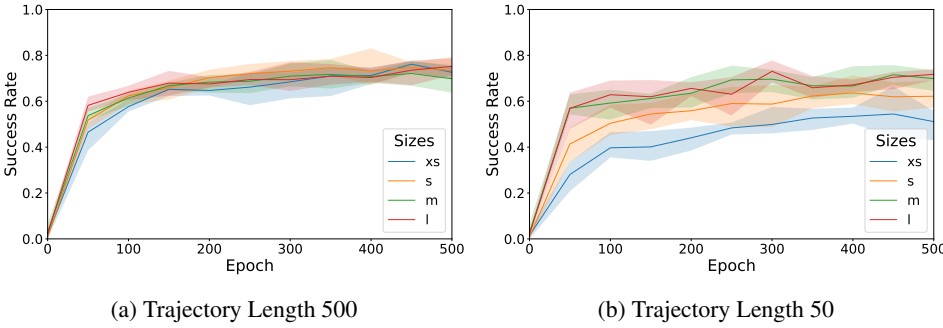

(a) Trajectory Length 500

(b) Trajectory Length 50

Figure 8: **Effect of VAE decoder size**: For longer trajectories, even the smallest decoder, $xs$, is sufficient to give high task performance. For shorter trajectories, a larger decoder model helps maintain the same level of performance.

| Policy | MLP 512 | MLP 256 | MLP 128 | DP (large) | DP (small) | LWD |
|---|---|---|---|---|---|---|
| **# Parameters** | 1396.2k | 370.4k | 103.3k | 2086.88k | 403.636k | 77.1k |
| button press | 1.00 | 1.00 | 1.00 | 1.00 | 1.00 | 1.00 |
| door open | 1.00 | 1.00 | 1.00 | 0.8 | 0.73 | 1.00 |
| drawer close | 1.00 | 0.87 | 0.87 | 1.00 | 0.6 | 1.00 |
| drawer open | 1.00 | 1.00 | 0.67 | 1.00 | 1.00 | 1.00 |
| peg insert | 0.13 | 0.20 | 0.27 | 0.0 | 0.0 | 0.53 |
| pick place | 0.13 | 0.00 | 0.00 | 0.2 | 0.0 | 0.13 |
| push | 0.13 | 0.00 | 0.13 | 0.4 | 0.13 | 0.47 |
| reach | 0.53 | 0.33 | 0.267 | 0.13 | 0.13 | 0.47 |
| window close | 1.00 | 1.00 | 1.00 | 1.0 | 0.93 | 1.00 |
| window open | 1.00 | 1.00 | 1.00 | 1.0 | 0.733 | 1.00 |
| **Mean over seeds** | 0.693 | 0.640 | 0.620 | 0.653 | 0.527 | 0.76 |
| **Stddev over seeds** | 0.072 | 0.04 | 0.061 | 0.011 | 0.064 | 0.052 |

Table 2: **Performance comparison across different tasks**. The baselines are three MLP policies, each with 5 hidden layers of the same size (512, 256, and 128 neurons), and two diffusion policy models with comparable sizes (denoted as DP(large) and DP(small)).

### 4.3 VAE DECODER SIZE ABLATION

As noted in subsection 4.2, the size of the hypernetwork decoder influences the quality of decoded policies for the MT10 task suite, when trained on trajectory snippets. Here we conduct an ablation on the decoder size, evaluating the average success rate of decoded policies across all MT10 tasks. Figure 8 illustrates the performance of decoders with varying sizes, denoted as $xs$ (3.9M parameters), $s$ (7.8M parameters), $m$ (15.6M parameters), and $l$ (31.2M parameters). It's important to note that despite the substantial parameter count of the hypernetwork decoder, the resulting inferred policy remains relatively small ($< 100K$ parameters, see Table 2). The results demonstrate that increasing the decoder size consistently improves the average success rate of the decoded policies. More details regarding the decoder size characterization is provided in Appendix C.1

This contrasts with the observations from the HalfCheetah environment, where even smaller decoders generated accurate policies from trajectory snippets. We hypothesize that this discrepancy stems from two key factors. First, the cyclic nature of HalfCheetah provides sufficient information within the snippets to infer the underlying policy. Second, the increased complexity of the MT10 tasks means that snippets may lack crucial information for policy inference. For instance, in a pick-and-place task, a snippet might only capture the "pick" action, leaving the latent representation without sufficient information to infer the "place" action.

### 4.4 METAWORLD MT10 SIZE BENCHMARK

In the previous experiments, we have shown that LWD can learn a distribution of policies from a trajectory dataset. However, another common use case for robotic learning is multi-task imitation learning. In this experiment, we study the ability of LWD to learn a task conditional distribution of policies. We test the performance of LWD on the Metaworld MT10 benchmark and compare it to a vanilla multi-task MLP policy trained on the same dataset. The baselines are three MLP policies, each with 5 hidden layers of the same size (512, 256 and 128 perceptions), and two diffusion policy models with a comparable number of parameters. Appendix C.6 shows the implementation details of the diffusion policy model.. All baseline models have the task identifier

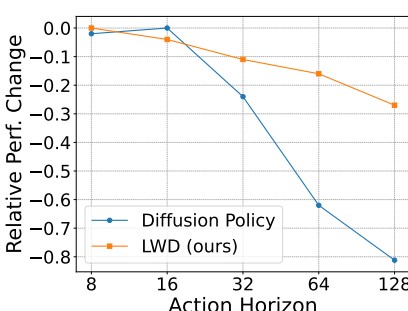

Figure 9: LWD maintains performance for longer action horizons, allowing fewer diffusion queries for the same episode length.

appended to the state input. As seen in Table 2, LWD outperforms the vanilla multi-task policy in the average success rate across all tasks while having fewer parameters during inference. We also see that the policy generated by LWD has a smaller parameter count than the vanilla multi-task policy,

with about 1/18th the number of parameters. Further, we see that the diffusion policy model also struggles to achieve high multi-task success rates even with about 27 times the number of parameters of LWD's generated policy.

### 4.5 EFFECT OF ACTION HORIZON

The previous experiments have shown that LWD can learn a distribution of behaviors for a task or a distribution of policies for a set of tasks. All these experiments were conducted with a single policy being generated at the start of the episode. However, for environments where the training dataset has a high variance, it is important to diffuse out locally optimal policies. That is, we need to diffuse a policy out every $H_a$ steps, where $H_a$ is the action horizon. This is what we observe for Diffusion Policy (Chi et al., 2024) as well. We use their PushT task, whose dataset has fewer trajectories with a higher variance. We compare the performance of LWD to Diffusion Policy on this task. Specifically, we compare the performance change as we change the action horizon from 8 to 128, relative to the best value. We now condition our latent diffusion model on the current state. We see in Figure 9 that Diffusion Policy had the best performance at a horizon of 16, whereas LWD had the best performance at a horizon of 8. But note that as the action horizon increases, the performance of Diffusion Policy degrades much faster than LWD. Therefore, for a relative performance reduction of $\sim 25\%$, LWD requires $1/4$th the number of expensive diffusion model queries.

## 5 LIMITATIONS AND FUTURE WORK

Although LWD has promising results, it has some limitations and avenues for future work. LWD needs a lot of trajectory data to learn the distribution of policies accurately. Moreover, by virtue of generating closed-loop policies, LWD is more prone to see out-of-distribution states when compared to methods that diffuse multi-step trajectories. These limitations serve as potential directions for future work. For example, it would be interesting to investigate methods to learn the distribution of policies from fewer demonstration trajectories. Another avenue is to use our method with a hypernetwork that can output weights for a transformer network, to take sequence data as input, or a vision transformer network, to take image data as input. Finally, performance of the policies could be improved by warm starting LWD with the previous solution or latent, to give the decoder more context.

## 6 CONCLUSION

We present LWD, a method to learn a distribution of policies from a heterogeneous set of demonstration trajectories. We first embed trajectories into a latent space and then learn the distribution of policies in this latent space. We then decode these latents to generate policies using a hypernetwork decoder. We show that LWD can reproduce the original policies present in the demonstration trajectories, in two cases. First, we show that we can reproduce multiple behaviors for the same task. We also show that we can reproduce policies used for multi-task learning. Finally, we discuss how LWD can be used with high-variance data, and compare it to baselines. We believe that LWD can be a useful tool for learning from demonstration, and can be used in a variety of applications, including robotics, reinforcement learning, and imitation learning.

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

## A   APPENDIX – MODIFIED ELBO DEREVATION

We derive a modified version of the Evidence Lower Bound (ELBO) to incorporate $p(\theta)$. This is shown below.

$$\log p(\tau) = \log \int \int p(\tau, \theta, z) \, dz \, d\theta \quad \text{(Introduce policy parameter } \theta \text{ and latent variable } z)$$

$$= \log \int \int p(\tau \mid z, \theta) p(\theta \mid z) p(z) \, dz \, d\theta \quad \text{(Apply the chain rule)}$$

$$= \log \int \int \frac{p(\tau \mid z, \theta) p(\theta \mid z) p(z)}{q(z \mid \tau)} q(z \mid \tau) \, dz \, d\theta \tag{5a}$$

$$\text{(Introduce a variational distribution } q(z \mid \tau) \text{, approximating the true posterior } p(z \mid \tau))$$

$$= \log \int \mathbb{E}_{p(\theta \mid z)} \left[ \frac{p(\tau \mid z, \theta) p(z)}{q(z \mid \tau)} q(z \mid \tau) \right] dz \tag{5b}$$

$$= \log \mathbb{E}_{q(z \mid \tau)} \left[ \frac{\mathbb{E}_{p(\theta \mid z)} \left[ p(\tau \mid z, \theta) \right] p(z)}{q(z \mid \tau)} \right] \tag{5c}$$

$$\geq \mathbb{E}_{q(z \mid \tau)} \left[ \log \left( \frac{\mathbb{E}_{p(\theta \mid z)} \left[ p(\tau \mid z, \theta) \right] p(z)}{q(z \mid \tau)} \right) \right] \quad \text{(Jensen's inequality)}$$

$$= \mathbb{E}_{q(z \mid \tau)} \left[ \log \left( \mathbb{E}_{p(\theta \mid z)} \left[ p(\tau \mid z, \theta) \right] \right) \right] - \mathbb{E}_{q(z \mid \tau)} \left[ \log \left( q(z \mid \tau) \right) - \log \left( p(z) \right) \right] \tag{5d}$$

$$= \mathbb{E}_{q(z \mid \tau)} \left[ \log \left( \mathbb{E}_{p(\theta \mid z)} \left[ p(\tau \mid \theta) \right] \right) \right] - \text{KL}(q(z \mid \tau) \parallel p(z)) \quad \text{(conditional independence)} \tag{5e}$$

$$\geq \mathbb{E}_{q(z \mid \tau)} \left[ \mathbb{E}_{p(\theta \mid z)} \left[ \log \left( p(\tau \mid \theta) \right) \right] \right] - \text{KL}(q(z \mid \tau) \parallel p(z)) \quad \text{(Jensen's inequality)} \tag{5f}$$

Assuming the state transitions are Markov and $s_1$ is independent of $\theta$, the joint likelihood of the entire sequence $\{(s_1, a_1), (s_2, a_2), \ldots, (s_T, a_T)\}$ (i.e., $p(\tau \mid \theta)$) is given by:

$$p(s_1, a_1, \ldots, s_T, a_T \mid \theta) = p(s_1) p(a_1 \mid s_1, \theta) \cdot \prod_{t=2}^{T} p(s_t \mid s_{t-1}, a_{t-1}) p(a_t \mid s_t, \theta) \tag{6a}$$

$$\log p(s_1, a_1, \ldots, s_T, a_T \mid \theta) = \log p(s_1) + \log p(a_1 \mid s_1, \theta) \tag{6b}$$

$$+ \sum_{t=2}^{T} \left[ \log p(s_t \mid s_{t-1}, a_{t-1}) + \log p(a_t \mid s_t, \theta) \right] \tag{6c}$$

$$= \sum_{t=1}^{T} \left[ \log p(a_t \mid s_t, \theta) \right] + A \tag{6d}$$

$A$ are all the terms that do not contain $\theta$. Substituting $2d$ in $1f$:

$$\log p(\tau) \geq \mathbb{E}_{q(z \mid \tau)} \left[ \mathbb{E}_{p(\theta \mid z)} \left[ \log \left( p(\tau \mid \theta) \right) \right] \right] - \text{KL}(q(z \mid \tau) \parallel p(z))$$

$$= \mathbb{E}_{q(z \mid \tau)} \left[ \mathbb{E}_{p(\theta \mid z)} \left[ \sum_{t=1}^{T} \left[ \log p(a_t \mid s_t, \theta) \right] \right] \right] + A - \text{KL}(q(z \mid \tau) \parallel p(z)) \tag{7}$$

We ignore the terms in $A$ as these cannot be subject to maximization as we do not have access to the state transition probabilities (although this can be modeled with a world model, we leave it as a direction for future work).

Therefore, our modified ELBO is:

$$mELBO = \sum_{t=1}^{T} \left[ \mathbb{E}_{q(z \mid \tau)} \left[ \mathbb{E}_{p(\theta \mid z)} \left[ \log p(a_t \mid s_t, \theta) \right] \right] \right] - \text{KL}(q(z \mid \tau) \parallel p(z)) \tag{8}$$

# B   APPENDIX C – VAE LOSS DERIVATION

Since $a_t \sim \mathcal{N}(\pi(s_t, \theta), \sigma^2)$:

$$p(a_t \mid s_t, \theta) = \frac{1}{\sqrt{2\pi\sigma^2}} \exp\left(-\frac{(a_t - \pi(s_t, \theta))^2}{2\sigma^2}\right) \tag{9}$$

Our objective is to maximize the $mELBO$. The likelihood of trajectory $\tau_k = \{s_t^k, a_t^k\}_{t=1}^T$ for the given VAE parameters is:

$$\begin{aligned}
\mathcal{L}\left(\{s_t^k, a_t^k\}_{t=1}^T \mid \phi_{enc}, \phi_{dec}\right) &= \sum_{t=1}^T \mathbb{E}_{q_{\phi_{enc}}(z|\{s_t^k, a_t^k\}_{t=1}^T)} \left[\mathbb{E}_{p_{\phi_{dec}}(\theta|z)} \left[\log p\left(a_t^k \mid s_t^k, \theta\right)\right]\right] \\
&\quad - \text{KL}\left(q_{\phi_{enc}}\left(z \mid \{s_t^k, a_t^k\}_{t=1}^T\right) \| p(z)\right) \\
&= C - \frac{1}{2\sigma^2} \sum_{t=1}^T \mathbb{E}_{q_{\phi_{enc}}(z|\{s_t^k, a_t^k\}_{t=1}^T)} \left[\mathbb{E}_{p_{\phi_{dec}}(\theta|z)} \left[(a_t^k - \pi(s_t^k, \theta))^2\right]\right] \\
&\quad - \text{KL}\left(q_{\phi_{enc}}\left(z \mid \{s_t^k, a_t^k\}_{t=1}^T\right) \| p(z)\right) \tag{10}
\end{aligned}$$

For computational stability, we construct our decoder to be a deterministic function $f_{\phi_{dec}}$, i.e., $p_{\phi_{dec}}(\theta \mid z)$ becomes $\delta(\theta - f_{\phi_{dec}}(z))$, therefore:

$$\begin{aligned}
\mathcal{L}\left(\{s_t^k, a_t^k\}_{t=1}^T \mid \phi_{enc}, \phi_{dec}\right) &= C - \frac{1}{2\sigma^2} \sum_{t=1}^T \mathbb{E}_{q_{\phi_{enc}}(z|\{s_t^k, a_t^k\}_{t=1}^T)} \left[(a_t^k - \pi(s_t^k, f_{\phi_{dec}}(z)))^2\right] \\
&\quad - \text{KL}\left(q_{\phi_{enc}}\left(z \mid \{s_t^k, a_t^k\}_{t=1}^T\right) \| p(z)\right)
\end{aligned}$$

Enforcing $p(z) = \mathcal{N}(0, I)$, and ignoring constants unaffected by the VAE parameters, we get:

$$\begin{aligned}
\mathcal{L}\left(\{s_t^k, a_t^k\}_{t=1}^T \mid \phi_{enc}, \phi_{dec}\right) &= -\sum_{t=1}^T \mathbb{E}_{q_{\phi_{enc}}(z|\{s_t^k, a_t^k\}_{t=1}^T)} \left[(a_t^k - \pi(s_t^k, f_{\phi_{dec}}(z)))^2\right] \\
&\quad - \beta_{kl} \sum_{i=1}^{\dim(z)} \left(\sigma_i^2 + \mu_i^2 - 1 - \log \sigma_i^2\right) \tag{11}
\end{aligned}$$

where, $(\mu, \sigma) = f_{\phi_{enc}}(\{s_t^k, a_t^k\}_{t=1}^T)$, $z \sim \mathcal{N}(\mu, \sigma)$, and $\beta_{kl}$ is the regularization weight.

# C   APPENDIX – IMPLEMENTATION DETAILS

The following are the hyperparameters we use for our experiments:

## C.1   VAE HYPERNETWORK DECODER SIZE CHARATERIZATION

For the hyper network, we utilize an HMLP model from the https://hypnettorch.readthedocs.io/en/latest/ package with default parameters. To vary the size of the decoder, as explained in subsection 4.3, we set the hyperparameter in the HMLP as shown in Table 3

## C.2   DIFFUSION MODEL PARAMETERS

For all our experiments, the diffusion model hyperparameters are the same, as given below. We use the model described (with its default hyperparameters overridden with the parameters given below) in https://nn.labml.ai/diffusion/ddpm/index.html.

| Size | No. of parameters | layers |
|------|-------------------|--------|
| xs | 3.9M | [50, 50] |
| s | 7.8M | [100, 100] |
| m | 15.6 M | [200, 200] |
| l | 31.2M | [400, 400] |

Table 3: VAE size varying parameters

| Parameter | Value |
|-----------|-------|
| Diffusion Learning Rate | $3 \times 10^{-4}$ |
| Conditioning Dimension | 4 |
| Channels | 32 |
| Attention Levels | $\{0, 1\}$ |
| Number of Residual Blocks | 1 |
| Channel Multipliers | $\{1, 2, 4\}$ |
| Number of Heads | 2 |
| Transformer Layers | 1 |
| Number of diffusion steps | 1000 |

Table 4: Diffusion model hyperparameters

| Parameter | Value |
|-----------|-------|
| Trajectory Length | 1000 |
| Batch Size | 32 |
| VAE Num Epochs | 150 |
| VAE Latent Dimension | 256 |
| VAE Decoder Size | `l` |
| Evaluation MLP Layers | $\{256, 256\}$ |
| VAE Learning Rate | $3 \times 10^{-4}$ |
| KL Coefficient | $1 \times 10^{-6}$ |
| Diffusion Num Epochs | 200 |

Table 5: Mujoco locomotion hyperparameters.

## C.3 MUJOCO LOCOMOTION TASKS

We use the following hyperparameters to train VAEs for all D4RL mujoco tasks shown in the paper. To show the effect of shorter trajectories in subsection 4.2, we change the Trajectory Length to 100.

## C.4 ADROIT HAMMER TASK

We use the same hyperparameters as Table 5 and override the following hyperparameters to train VAEs for the D4RL Adroit hammer task shown in the paper.

| Parameter | Value |
|-----------|-------|
| Trajectory Length | 128 |
| VAE Num Epochs | 20 |
| Diffusion Num Epochs | 10 |

Table 6: Adroit hammer hyperparameters.

Further, for the experiment where we show the hammer task can be composed of sub-tasks, we change the Trajectory Length to 32 to enable LWD to learn the distribution of shorter horizon policies.

## C.5 METAWORLD MT10 TASKS

For all the experiments shown in subsection 4.4, we use the same hyper-parameters described in Table 5, and override the following:

| Parameter | Value |
|---|---|
| Trajectory Length | 500 |
| VAE Num Epochs | 100 |
| Diffusion Num Epochs | 100 |
| VAE Decoder Size | xs |

Table 7: MT10 hyperparameters.

To show the effect of shorter trajectories in subsection 4.2, we change the Trajectory Length to 50.

## C.6 METAWORLD MT10 DIFFUSION POLICY MODEL

To train the diffusion policy baseline model shown in Table 2, we utilize the training script provided by the authors of DP here:

https://colab.research.google.com/drive/1gxdkgRVfM55zihY9TFLja97cSVZOZq2B?usp=sharing.

To set the model size we use the following parameters:

| Size | Diffusion Step Embed Dim | Down Dims | Kernel Size |
|---|---|---|---|
| small | 64 | [16, 32, 64] | 5 |
| large | 256 | [32, 64, 128] | 5 |

Table 8: Details for DP implementation.

# D LATENT REPRESENTATIONS

## D.1 MUJOCO HALFCHEETAH

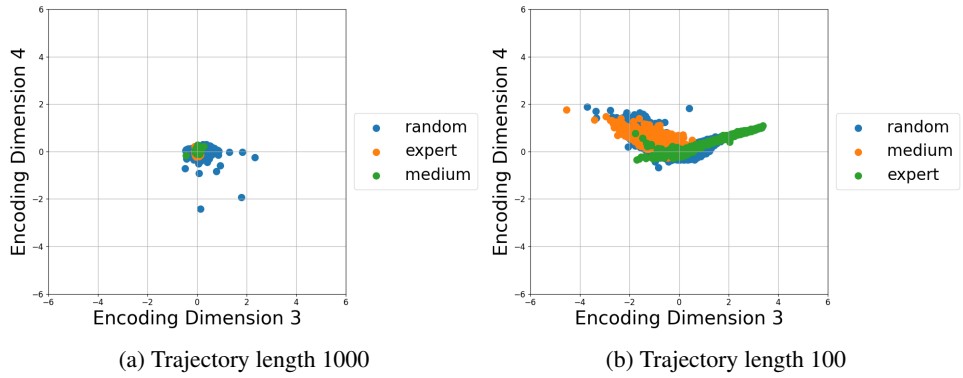

(a) Trajectory length 1000      (b) Trajectory length 100

Figure 10: **Effect of trajectory snipping** in HalfCheetah. Top third and fourth principal components of the latent.

## D.2 MT10

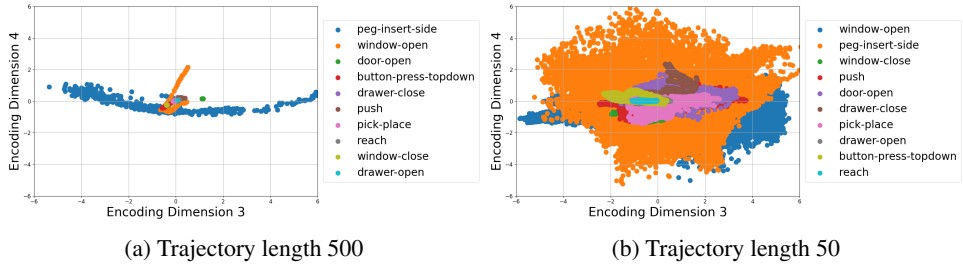

(a) Trajectory length 500

(b) Trajectory length 50

Figure 11: **Effect of trajectory snipping** in MT10. Top third and fourth principal components of the latent.

