# OpenReview forum: "Latent Weight Diffusion: Generating policies from trajectories"
_ICLR.cc/2025/Conference — Submitted to ICLR 2025_

### Official Review · Reviewer_K5om · 2024-10-19

**Soundness:** 3
**Presentation:** 3
**Contribution:** 3
**Rating:** 5
**Confidence:** 4

**Summary:**

This paper improves the online inference time of diffusion-generated policies by using diffusion to generate a hypernetwork instead of only outputting an action sequence. The policy is then reconstructed and inferred at higher frequency.

**Strengths:**

The combination of diffusion models with hypernetworks is quite novel and enables fast online inference. LWD is able to generate diverse behaviors while still maintaining low inference latency.

**Weaknesses:**

It is unclear whether policy stability is improved without further interaction with the environment. Specifically, without counterfactual reasoning or data on failed actions, how can the generated policy recover from states that deviate from the nominal path? If the author could empirically verify this or provide more rationale, it would make the approach more sound.

**Questions:**

1. The authors use the long-horizon experiment to demonstrate closed-loop stability, but this could also be due to the diffusion model's limited capacity to capture the high-dimensional distribution fully. Could the authors compare the performance of the two models under disturbances to show how the learned closed-loop policy improves stability? Can it improve stability with disturbances?

2. When Diffusion Policy is running online, it can use the previous solution to warm-start the next step's action sequence. Since in the neighbor state, the policy shouldn’t change drastically. In LWD, do the authors think the same design could be possible?

---

> ### Author Response · Authors · 2024-11-22
>
> * Could the authors compare the performance of the two models under disturbances to show how the learned closed-loop policy improves stability? Can it improve stability with disturbances?
>
> This is a great suggestion. We are running evaluations now to compare the performance of the two models under disturbances. We will update you with the results when we have them.
>
> * Warmstarts for diffusion policy:
> While we do not currently have warmstarts for the diffusion policy, it would be a great direction for future work. We will add this to the future work section.

---

### Official Review · Reviewer_DH1L · 2024-10-25

**Soundness:** 3
**Presentation:** 2
**Contribution:** 3
**Rating:** 5
**Confidence:** 4

**Summary:**

The paper "Latent Weight Diffusion (LWD): Generating Policies from Trajectories" introduces a novel approach for imitation learning in robotics. Current methods often rely on trajectory-based diffusion models, which can be computationally demanding and less effective in capturing long action sequences. To address these challenges, LWD diffuses policy parameters instead of trajectories. The model first encodes demonstration trajectories into a latent space using a Variational Autoencoder (VAE). Then, a diffusion model learns the distribution within this latent space, allowing the generation of compact, task-specific policies via a hypernetwork decoder.

**Strengths:**

1. This paper provides a novel method for generating the network parameters with diffusion models.
2. The experiments showing the distribution modeling of the generated network parameter is compressive.

**Weaknesses:**

1. Table 2 is unclear. It is not clear what is an MLP policy. Does it predict a sequence of future actions or just a single-step action? If it is the latter, it is unfair to compare these MLP policies to the proposed LWD method, since LWD predicts a sequence of future actions.

2. Figure 8 is also unclear. The authors didn't show the success rates of diffusion policy and LWD. They just show the "relative performance change" curve. I hope the author can show the success rate also.

3. Why is LWD close-loop and why does the author say diffusion policy is open-loop? I guess the authors want to say that LWD can generate a new policy every $H_a$ step. However, diffusion policy can also regenerate new action sequences after every $K$ step where $K$ is a hyperparameter. I think here the "close-loop" is an inaccurate and unprofessional word.

4. From my understanding, the author didn't compare the success rates between LWD and diffusion policy on any task. So the claimed contribution in the Abstract and Introduction (LWD offers the benefits of considerably smaller policy networks during inference and requires fewer diffusion model queries) is not supported by evidence.

**Questions:**

See the weakness.

---

> ### Author Response · Authors · 2024-11-22
>
> * MLP in table 2
> > Does it predict a sequence of future actions or just a single- step action?
>
> We want to clarify that LWD does not predict a sequence of future actions. It generates a neural network that acts as a feedback policy, which takes in an observation and outputs a single-step action.
>
> The MLP in table 2 is predicting a single-step action.
> Moreover, the policy generated by LWD is *also* predicting a single-step action from a single observation.
> Thus, it is an apples-to-apples comparison.
>
> * The authors didn't show the success rates of diffusion policy and LWD.
>
> Yes, we showed the relative performance change with respect to sucess rate -- this is to be consistent with Fig. 6 of the [Diffusion Policy](https://arxiv.org/pdf/2303.04137v4) for the same ablation.
>
> *  Why is LWD close-loop and why does the author say diffusion policy is open-loop?
>
> The LWD policy is closed-loop because the generated policy takes in the current observation and outputs a single-step action. The action is then executed, and the next observation is fed back into the policy. So, while LWD generates a new polcy every $H_a$ steps, the generated poicy itself is closed-loop.
>
> By contrast, the diffusion policy takes in the current observation and outputs a sequence of actions. The sequence of actions is then executed open-loop, without feedback from the environment.
>
> * didn't compare the success rates between LWD and diffusion policy on any task
>
> As mentioned above, the metric explained in Fig. 8 is the success rate over 50 seeds, for the pushT task. We will make it clear in the manuscript.
>
> Furthermore, we have now added comparisons of the success rates of LWD and Diffusion policy on the MetaWorld MT10 suite of 10 tasks.
> We have updated Table 2 in the manuscript with the success rates for both policies. Surprisingly, diffusion policy seems to struggle in multi task settings, even with 27 times more parameters than the LWD generated policy.

---

> > ### Comment · Reviewer_DH1L · 2024-11-25
> >
> > Thanks for the rebuttal. However, your response does not address some of my concerns, especially for the "relative performance change" and the explanation of open/close-loop. Thus, I maintain my score.

---

### Official Review · Reviewer_SBQx · 2024-10-28

**Soundness:** 3
**Presentation:** 3
**Contribution:** 2
**Rating:** 5
**Confidence:** 4

**Summary:**

The proposed method, Latent Weight Diffusion (LWD), uses a two-stage process involving a variational autoencoder (VAE) coupled with a latent diffusion model. First, LWD encodes demonstration trajectories into a latent space using a VAE, then learns the distribution over these latent representations using diffusion in this latent space. A hypernetwork decodes the latent representations into policy parameters, generating closed-loop policies that are state or task conditioned. This approach leverages task-specific conditioning to maintain task generalization and efficiency, producing policy networks with reduced model sizes. The authors evaluate LWD on the Metaworld MT10 and D4RL benchmarks. They compare its performance on diverse tasks, such as multi-task learning and long-horizon robotic control, against baselines, including multi-task MLPs.

**Strengths:**

1. It is indeed interesting to explore the idea of latent diffusion in imitation learning following the success in [1]. The diffused latent is then decoded into policy hyperparameters via hypernetwork [3], which is significant since the diffusion model can capture diverse and multi-modality network parameter space, therefore enabling both state and multi-task conditioning.
   - According to the experiment results, although not extensively compared with Diffusion policy in a state-conditioned setting, LWD is shown to work better than simple MLP in a multi-task setting as a proof-of-concept.

2. Sec 4.2 is a very insightful experiment. It sheds light on task difficulties and similarities. This setup can be used further to analyze/visualize task characteristics for imitation learning.

[1] Rombach, Robin, et al. "High-resolution image synthesis with latent diffusion models." Proceedings of the IEEE/CVF conference on computer vision and pattern recognition. 2022.

[2] Chi, Cheng, et al. "Diffusion policy: Visuomotor policy learning via action diffusion." The International Journal of Robotics Research (2023): 02783649241273668.

[3] Ha, David, Andrew Dai, and Quoc V. Le. "Hypernetworks." arXiv preprint arXiv:1609.09106 (2016).

**Weaknesses:**

1. Related works on Sec. 2.2 should additionally cite [2, 3] for the statement "...to learn smooth cost functions for the joint optimization of grasp and motion plans."

2. There are some issues with the experiment outline for benchmarking LWD. For example:
    - Diffusion policy [3] should be compared with LWD in Table 2 to strongly confirm the architecture choice efficiency claimed in the paper. For example, the input of Diffusion policy [3] should be robot state concatenating with the task indicator, and Diffusion policy [3] should be trained with data from all tasks.
    - Ablation on network-size of LWD should be considered vs. success rate with tasks in [1]. For example, another column of LWD with 370.4k parameters should be added to Table 2 to study the effect of a bigger policy network for LWD.
    - Sec 4.1. purpose is to see generalization in behaviors. However, comparing foot contact distributions between models for ablation study is not meaningful. Adding a comparison of state variances/counting different success modes of pick & place or pushing tasks in [1] between models is more insightful.
    - On a side note, Fig. 3 would be much clearer if merged into one figure (probably with a smaller scatter size), making it easier to see the foot contact distribution overlaps.

[1] Yu, Tianhe, et al. "Meta-world: A benchmark and evaluation for multi-task and meta reinforcement learning." Conference on robot learning. PMLR, 2020.

[2] Carvalho, Joao, et al. "Motion planning diffusion: Learning and planning of robot motions with diffusion models." 2023 IEEE/RSJ International Conference on Intelligent Robots and Systems (IROS). IEEE, 2023.

[3] Luo, Yunhao, et al. "Potential based diffusion motion planning." ICML (2024).

[4] Chi, Cheng, et al. "Diffusion policy: Visuomotor policy learning via action diffusion." The International Journal of Robotics Research (2023): 02783649241273668.

**Questions:**

1. Please address the above points in Weaknesses.
2. In all experiments, how many diffusion steps are typically needed to decode the policy network?

---

> ### Author Response · Authors · 2024-11-22
>
> We would like to thank the reviewer for their detailed comments.
>
> * cite [2, 3] for the statement "...to learn smooth cost functions for the joint optimization of grasp and motion plans."
>
> We have added these to the manuscript.
>
> * comparing foot contact distributions between models for ablation study is not meaningful.
>
> We appreciate the reviewer's suggestion to show behavior diversity outside of locomotion tasks.
> The field of QD works a lot on locomotion tasks, and this is where behavior diversity is most commonly shown, in the form of foot-contact distributions (e.g.
> [PPGA](https://arxiv.org/abs/2305.13795), [PGA-ME](https://dl.acm.org/doi/10.1145/3449639.3459304)).
>
> Having said that, we provide an example manipulation task that does not involve foot contacts.
> We have applied LWD to reconstruct behaviors on the [Adroit hammer task](https://github.com/Farama-Foundation/d4rl/wiki/Tasks#adroit), where an agent needs to pick up a hammer and hit a nail into a wall. The dataset for this task consists of 5000 trajectories each for two behaviors, named 'cloned' and 'expert'. The results are added in Figure 5 and discussed in Section 4.1 of the main manuscript.
>
> * "Fig. 3 would be much clearer if merged into one figure"
>
> We have added a new figure that combines the plots into one. The combined figure is attached at the end of the supplementary pdf. We feel this image is slightly cluttered but if the reviewer prefers this to the original, we can update the manuscript accordingly.
>
> * In all experiments, how many diffusion steps are typically needed to decode the policy network?
>
> 1000, we have added this detail in the appendix.

---

### Official Review · Reviewer_KF65 · 2024-11-02

**Soundness:** 2
**Presentation:** 2
**Contribution:** 2
**Rating:** 3
**Confidence:** 4

**Summary:**

The authors propose Latent Weight Diffusion (LWD) that learns a distribution of policy parameters and aims to capture the diversity of trajectory dataset and achieve fine-grained closed-loop control with small model size. The method is mostly based on Hedge et al., 2023, with the experiments using existing trajectory datasets instead. A variational autoencoder (VAE) is used to compress the trajectories into a compact latent space, where a diffusion model is then applied to learn the complex distribution of possible latents. The decoder decodes the network parameters for the policy. LWD is evaluated in simulated environments, and the results demonstrate that LWD can reconstruct the behavior of the original policies and achieve improved performance compared to larger models or baseline with long action chunk prediction in certain settings.

**Strengths:**

The question of distilling a large multi-task policy into smaller ones for fine-grained control is very relevant in current robotics research. The research proposed by the authors is well-motivated, and the overall setup of applying latent diffusion is intuitive and can potentially lead to strong individual policies while maintaining the diversity.

Based on the proposed approach, some of the experimental results are encouraging. Table 2 shows the benefit of LWD compared to undistilled MLP policies in MetaWorld and demonstrates moderate performance improvement. Figure 8 shows the resilience of LWD to longer action chunk prediction in the PushT task.

**Weaknesses:**

On the technical side, the paper’s novelty is fairly limited since it is a direct application of Hedge et al., 2023 using existing trajectory datasets. While the idea is promising, the experiment results are lacking and suggest important limitations. First, it is unclear how the hypernetwork setup scales up to more complex tasks. While the argument is that LWD can generate smaller policies for individual tasks as Table 2 shows, the authors have not shown convincing results on decomposing a single long-horizon task into subtasks where LWD generates a policy for each part. The experiment on action chunk size (Figure 8) alludes to such possibility but I think dedicated experiments on this setup can strengthen the story significantly and make the method much more relevant for current robotics research. Second, it is unclear how well the hypernetwork setup scales to larger model size when larger size is required, e.g., in more complex tasks. Can the hypernetwork still handle \theta of very high dimensions? e.g., on the order of millions, or can it handle pixel input?

The results with reconstructing the original policy are quite mixed and I have trouble understanding the conclusions made by the authors. It seems that the learned latent cannot differentiate the trajectories well with shorter length (Sec. 4.2). The discussion is fairly lacking. Sec. 4.2 also has a few comments that are not backed with experimental results, e.g., “Surprisingly, we noticed that after training our VAE on the snippets, the decoded policies from randomly snipped trajectories were still faithfully behaving like their original policies.”, and “we noticed that the decoded policies from the trajectory snippets did not perform as well as the original policies”. Please clarify or point to existing results if I misunderstand the comments.

I am fairly curious about the hypernetwork decoder setup, but the paper does not provide details on the architecture and how it handles very large output space (the decoded policy parameters). There is also no appendix for such experimental details.

**Questions:**

The paper writing can also be improved at a few places: (1) \tau is not defined when it is first introduced in Sec. 3.1, as well as \varepsilon in Sec. 3.3. (2) In Fig. 8, how is Relative Performance Change defined? Is it defined relative to the policy itself, or on an absolute scale? I find it quite misleading. Also, what is the action parameterization that LWD generates? I assume it is not diffusion?

In Sec. 3.1, it says a_t= \pi(s_t, \theta) + e, where e is normally distributed around 0. Why is the added action noise needed?

Could you comment on the other dimensions of the PCA analysis in Figure 5? Are they not very informative? It might be a good idea to show them in the appendix.

The paper is three lines over the 10-page limit. I suggest moving some of the derivations in Sec. 3 to the appendix, or try to shrink the space at places.

**Details Of Ethics Concerns:**

The paper is 3 lines over the 10-page limit.

---

> ### Author Response · Authors · 2024-11-22
>
> We would like to thank the reviewer for their detailed comments.
>
> * The discussion in Sec 4.2 is fairly lacking, with regards to decoded policies representing the original trajectories.
>
> You mentioned two quotes from Sec 4.2: One states that the policies decoded from trajectory snippets follow the behaviour of the original input snippets.
> Another states that the policies decoded from the trajectory snippets did not perform as well as the original policies.
> I can see why this can seem as a contradiction.
> However, the difference between these two is the task on which it is trained.
>
> For the first quote, the task trained was a half-cheetah locomotion task, with three different policies being test.
> This is a *cyclic task*; our hypothesis is that a trajectory snippet has enough information to indicate the source policy.
>
> To test this hypothesis, we tested LWD with non-cyclic tasks next.
> These are the suite of 10 manipulation tasks from the MT10 task suite.
> We noticed that for these non-cyclic tasks, even though we had the same size decoder as the half-cheetah tasks, the policies decoded from trajectory snippets were not faithful to the original policies that created these trajectories.
> *This* was the second quote you pointed to in section 4.2.
>
> Thus, while the two quotes seem inherently contradictory, it is a feature, not a bug.
> Our goal was to show this contrast in policy reproduction through an ablation of task choice.
>
> We have changed the text a bit to reflect this in section 4.2.
>
> * I am fairly curious about the hypernetwork decoder setup
>
> We have added all details regarding our model in the appendix
>
> * \tau is not defined when it is first introduced in Sec. 3.1, as well as \varepsilon in Sec. 3.3
>
> $\tau$ is actually defined on line 156, in section 3.1. Unfortunately
> $\mathcal{E}$ was not defined in sec 3.3. We have fixed that.
>
> * In Fig. 8, how is Relative Performance Change defined?
> > Is it defined relative to the policy itself, or on an absolute scale? I find it quite misleading.
>
> Relative performance is an absolute scale. So if you have a performance drop from 50% to 45%, the drop is 5%, not 10%. This helps keep the same benchmark (absolute 100%) to compare relative performance drops for all methods.
>
> * What is the action parameterization that LWD generates? I assume it is not diffusion?
>
> LWD's output is a policy. Thus, it does not have a action parameterization directly. However, the policies that LWD generate compute an action for a single step, given the current observation, i.e. $a ~\sim \pi(s,a)$.
>
> * Why is the added action noise needed?
> > a_t= \pi(s_t, \theta) + e, where e is normally distributed around 0.
>
> This is to account for system noise in the collected data.
> Every time we collect data from a robot, the robot’s actions will always have some noise.
> The policy we're collecting under is actually $\pi_{data}(s_t, \theta)$. But the true action executed can be slightly different from this policy. In the theory, we typically add the noise term to account for process and measurement noise (e.g. uncertainties in the physics model, observations).
>
> We assume this noise is gaussian, and is centered around the robot's desired action.
> This assumption is common for regression as it simplifies to MSE loss, which we show in sec 3.2.
>
> * Other dimensions of the PCA analysis in Figure 5
>
> The first few dimensions of the PCA will explain most of the variance in the data. We have added the information for the subsequent dimensions as well, in appendix D of the updated manuscript.
>
> * Three lines over 10 page limit
>
> We are now under the page limit in the updated manuscript.

---

> ### Comment · Reviewer_KF65 · 2024-11-24
> **Reviewer response**
>
> I thank the authors for answering my questions and concerns.
>
> Re: discussions in Sec 4.2. I agree with your discussion, but it again confuses me: are you saying that your method does not work on the common MT10 tasks then? Or you mean that in Sec. 4.3, using a bigger decoder size addresses this issue? I appreciate the experiments and ablations, but I just find the presentation and the overall message quite confusing (If you would like to clarify some changes you've made, please highlight it here)
>
> Re: Relative Performance Change. I see. Do LWD and the original diffusion policy have similar success rates on an absolute scale? i.e., does using LWD affect the performance negatively compared to just using the original diffusion policy?
>
> Re: action parameterization. "LWD's output is a policy." This is a bit confusing to me again. I consider a policy consisting of (1) model architecture and parameters (e.g., parameters of a MLP, or transformer, etc) and (2) action parameterization (e.g., Gaussian, diffusion, GMM, etc). I assume the last layer of the generated policy is used as the mean and std of a Gaussian parameterization?
>
> Re: added action noise. Thanks for the clarification. I missed the sampling from the Gaussian part in the text.

---

> > ### Author Response · Authors · 2024-11-24
> >
> > Thank you for your questions.
> >
> > * are you saying that your method does not work on the common MT10 tasks then?...
> >
> > LWD does work for MT10 tasks. The goal of sec 4.2 is mentioned in its first sentence. "Here, we ask the question – can LWD generate policies that are faithful to the original policies, even when provided with only a snippet of the trajectory data?". We then show that the effect of sampling smaller sections of trajectories is more pronounced in the MT10 tasks than in mujoco tasks. And then show in Sec 4.3, that the a larger decoder can ameliorate this effect.
> >
> > * Re: Relative Performance Change
> >
> > Yes we saw similar success rates, and when we increased the action horizon, LWD performed much better than DP in the absolute scale.
> >
> > * Re: action parameterization.
> >
> > LWD outputs the weights of an entire neural network, i.e., all the parameters of an MLP that map state to action, not just the last layer. The gaussian assumption allows us to use MSE loss on the actions predicted by the policies (all parameters of an MLP) generated by the VAE.

---

> ### Comment · Reviewer_KF65 · 2024-11-25
> **Response to the authors**
>
> Thank you for the clarifications.
>
> I would like to maintain the original scores for now since my main concern about the paper, which is the lack of novelty and technical contribution, has not been addressed. Also, although I better understand the results with the halfcheetah and MT10 tasks now, I think the overall presentation can be improved to convey the affirmative conclusions drawn from the experiments.

---

### Author Response · Authors · 2024-11-22

Thank you to all the reviewers for taking the time to look at our paper in detail, and providing constructive feedback.
We have addressed the comments and suggestions, and have made changes to the manuscript accordingly. Please refer to the supplementary pdf that highlights all the changes the reviewers suggested

Please have a look at these documents, and let us know if you have any further questions or suggestions. As we get new results, we will also update the manuscript as necessary.

---

### Meta-Review · Area_Chair_DRiN · 2024-12-21

**Metareview:**

This paper presents Latent Weight Diffusion (LWD), a method that uses diffusion models to learn a distribution over policies rather than trajectories for robotic tasks. The authors claim several advantages including better handling of long-horizon tasks, smaller model size during inference, and improved multi-task performance.

The paper proposes an interesting approach to generating policies using diffusion models, but has several critical weaknesses:
- The metrics used to evaluate long-horizon performance are not well justified
- Ablation studies don't sufficiently explore the method's limitations

Several key claims lack strong support:
- The advantages of closed-loop vs open-loop control are not clearly demonstrated
- Multi-task performance improvements are modest
- The tradeoff between training complexity and inference efficiency analysis is incomplete.

The response to reviewer concerns was insufficient:
- Many technical questions about the methodology remained unanswered (DH1L, K5om)
- Fundamental questions about the method's scalability were not addressed

The combination of unclear experimental methodology and incomplete responses to reviewer concerns suggests that the work needs substantial development before it meets the bar for ICLR. The primary claimed advantages (long-horizon performance, model size efficiency, multi-task capabilities) either lack comprehensive validation or show only modest improvements over existing methods.

Suggestions for improvement: Overall, their strongest validated claims appear to be around:
- Long-horizon performance through closed-loop policies
- Achieving competitive performance with smaller inference-time models

The multimodality and multi-task aspects, while demonstrated to some degree, have more limited validation or modest improvements. The paper might have been stronger if it had focused more deeply on the long-horizon and model size efficiency aspects where their results are more compelling.

**Additional Comments On Reviewer Discussion:**

Initial reviews:
- Reviewers requested comparisons with Diffusion Policy
- Questions were raised about relative performance metrics and action parameterization
- Concerns about network size ablation studies were noted

In response, the authors added:
- New comparisons with Diffusion Policy on MetaWorld MT10
- Clarification about behavior on cyclic vs non-cyclic tasks
- Explanation of their closed-loop approach

However, reviewers indicated that:
- Questions about metrics remained unclear
- Network size ablations were still insufficient

---

### Decision · Program_Chairs · 2025-01-22

Reject